# Inward- and outward-facing X-ray crystal structures of homodimeric P-glycoprotein CmABCB1

Atsushi Kodan[1], Tomohiro Yamaguchi[2], Toru Nakatsu[2,3], Keita Matsuoka[2], Yasuhisa Kimura[1], Kazumitsu Ueda[1,4] & Hiroaki Kato [2,3]

P-glycoprotein extrudes a large variety of xenobiotics from the cell, thereby protecting tissues from their toxic effects. The machinery underlying unidirectional multidrug pumping remains unknown, largely due to the lack of high-resolution structural information regarding the alternate conformational states of the molecule. Here we report a pair of structures of homodimeric P-glycoprotein: an outward-facing conformational state with bound nucleotide and an inward-facing apo state, at resolutions of 1.9 Å and 3.0 Å, respectively. Features that can be clearly visualized at this high resolution include ATP binding with octahedral coordination of $Mg^{2+}$; an inner chamber that significantly changes in volume with the aid of tight connections among transmembrane helices (TM) 1, 3, and 6; a glutamate—arginine interaction that stabilizes the outward-facing conformation; and extensive interactions between TM1 and TM3, a property that distinguishes multidrug transporters from floppases. These structural elements are proposed to participate in the mechanism of the transporter.

[1] Division of Applied Life Sciences, Graduate School of Agriculture, Kyoto University, Sakyo-ku, Kyoto 606-8502, Japan. [2] Department of Structural Biology, Graduate School of Pharmaceutical Sciences, Kyoto University, 46-29 Yoshida Shimoadachi-cho, Sakyo-ku, Kyoto 606-8501, Japan. [3] RIKEN Harima Institute at SPring-8, 1-1-1 Kouto, Sayo-cho, Sayo-gun, Hyogo 679-5148, Japan. [4] Institute for Integrated Cell-Material Sciences (WPI-iCeMS), Kyoto University, Sakyo-ku, Kyoto 606-8501, Japan. These authors contributed equally: Atsushi Kodan, Tomohiro Yamaguchi. Correspondence and requests for materials should be addressed to H.K. (email: katohiro@pharm.kyoto-u.ac.jp)

**P**-glycoprotein (P-gp; also known as ABCB1 or MDR1), a member of the ATP-binding cassette (ABC) transporter family, is a primary transporter that mediates active efflux of various hydrophobic chemicals from the cell[1–3]. Its substrates include more than a hundred chemically diverse molecules, including therapeutic drugs, steroid hormones, and signaling molecules[4]. P-gp plays important roles in normal physiology and is an essential component of many physiological barriers[5]. The protein consists of a minimum of four core domains: two transmembrane domains (TMDs), which create the translocation pathway for substrates, and two nucleotide-binding domains (NBDs), which bind and hydrolyze ATP to power the transport process[6]. These four domains can exist either as two separate polypeptides or fused together in a single long polypeptide with an internal duplication. Although several crystal structures of eukaryotic P-gp proteins have been determined in an inward-facing conformation[7–10], no outward-facing X-ray crystal structure was previously available, except for bacterial homologs[11,12], probably because the outward-facing state of P-gp in complex with a nucleotide is not stable enough to be crystallized[13]. The inward-facing structures reveal a large, kernel-shaped inner chamber in the center of TMDs, extending from the middle of the bilayer membrane to the cytosol[9]. This chamber is surrounded by all 12 TM helices and has two gates: one open to the membrane, and the other, located at the bottom center region of the TMDs, open to the cytosol[9]. At the apex of the chamber is a cluster of amino acids residues with aromatic hydrophobic side-chains, which are involved in substrate and inhibitor binding[7]. Recently, the overall conformational movement of mouse P-gp was determined by double electron−electron resonance[14]. However, the change in the structure of the inner chamber during substrate transport remained to be elucidated.

A recent study reported the cryo-EM structure of human P-gp in an outward-facing conformation[15]. The structure has only a small opening on the extracellular side of the membrane, and the TM helices are packed closely in the membrane inner leaflet. On the other hand, the outward-facing crystal structure of the bacterial homolog MsbA has wide gaps open to the outer leaflet region of the membrane; its large inner chamber, which reaches the intracellular region beyond the cell membrane, is exposed to the extracellular space and accessible from the outer leaflet[11,12]. The considerable difference between these structures may reflect differences in their functions (multidrug exporter vs. lipid flop-pase) or more general differences between eukaryotic and pro-karyotic ABC proteins. Due to the low resolution of the cryo-EM structure, however, it is difficult to investigate the detailed structural features, especially side-chain positions and the structural changes they undergo during the transport process via alternating conformational configurations[16].

Here we present a pair of crystal structures of dimeric P-gp from *Cyanidioschyzon merolae*, a transporter with quite similar functional properties to human P-gp[9]: an outward-facing nucleotide bound state (1.9 Å resolution) and an inward-facing apo state (3.0 Å). We solved these structures by generating a mutant in which the outward-facing state was sufficiently stable for crystallization; the same mutant was used to obtain both of the crystal structures. The resultant high-resolution crystallographic data are reliable enough to allow a discussion of structure and mechanism at the level of individual amino acid residues. Crucially, the conformational change from inward- to outward-facing is accompanied by shrinkage of the large inner chamber from the cytosolic side due to tilting and rotation of the TMDs. This contraction of the chamber could extrude a variety of substrates toward the extracellular side. The TMD motion is caused by NBD dimerization upon binding to $Mg^{2+}$ and ATP. NBD dimerization triggers a series of local conformational changes such as

movement of the Q-loop, intracellular helices IH1 and IH2, and TM helices via a relay of van der Waals interactions within the α3-Q-loop and coupling helices. Structural comparison of the two states suggests a free energy transduction process whereby the energy derived from ATP binding is stored in the structural strain of the TMDs.

## Results

**Structure determination.** To obtain the outward-facing conformation of CmABCB1 in a crystalline state, we introduced mutations that shifted the conformational equilibrium toward the outward-facing state in complex with $Mg^{2+}$ and nucleotide. Previously, we reported that the hydrogen bonds formed between TM1 and TM6 near the top of the inward-facing structure (Supplementary Fig. 1a) stabilize the inward-facing conformation in wild-type CmABCB1[9]. Accordingly, we generated a mutant in which these hydrogen bonds are disrupted by the Q147A^TM1/ T381A^TM6 double mutation (hereafter termed the QTA mutant). These mutations had a slight effect on protein function, as determined by drug resistance (Supplementary Fig. 1b) and ATP hydrolysis activity (Supplementary Fig. 1c–g, Supplementary Table 1). We successfully determined the outward-facing structure of the QTA mutant bound to the nonhydrolyzable ATP analog adenylyl imidodiphosphate (AMP-PNP) and $Mg^{2+}$ at a 1.9 Å resolution (Fig. 1c, d, g, h and Table 1). The extracellular hydrophilic region of the TMDs exhibited ambiguous electron density, suggesting that the upper parts of the TM helices and extracellular loops in this region are highly mobile. The inward-facing apo structure of the mutant, determined at a 3.0 Å resolution (Fig. 1a, b, e, f and Table 1), was almost identical to that of the WT[9], with a root-mean-squared deviation (r.m.s.d.) value of 0.47 Å for all Cα atoms. In the region containing the mutated Gln147^TM1 and Thr381^TM6 residues, the two structures are almost superimposable (Supplementary Fig. 1h).

Comparison of the inward- and outward-facing conformations of QTA CmABCB1 revealed that the overall conformational changes between both states is achieved by 9.5° tilting and 21° rigid-body rotation of the subunits (Supplementary Movie 1, 2). The helical interactions of TM6−TM4 and TM6−TM5 in the outward-facing structure are formed by a conformational change from the inward-facing structure (Fig. 1a, c), whereas the TM1 −TM6 interaction is dissociated (Fig. 1e, g). Interestingly, TM4 is partly unwound in the inward-facing conformation, but adopts a more regular alpha-helical geometry in the outward-facing conformation (Fig. 1e, g). Because of the conformational change, the large inner chamber with the ceiling in the inward-facing structure is significantly reduced in volume and open to the extracellular space in the outward-facing structure (Fig. 1b, d, f, h), suggesting that the large space in the inner chamber of the inward-facing structure varies due to conformational changes of the TMDs, like an elastically contractile machine. The QTA CmABCB1 structure in the outward-facing state is similar to that of human P-gp[15] but quite different from those of the bacterial homologs Sav1866[11] and MsbA[12] (Supplementary Fig. 2a). Each bacterial homolog still has a large inner space in the outward-facing conformation. In regard to the conformational change of mammalian P-gp, comparison among many inward-facing mouse P-gp structures revealed a tilting and rotation induced by NBD movement[10], similar to (or a component of) the conformational change in CmABCB1.

**Structural changes of the interior of the TMDs.** Interior views of the structures reveal local conformational changes combined with tilting and twisting of the TM helices (Fig. 2, Supplementary Movie 1), while lateral views of both dimer structures reveal

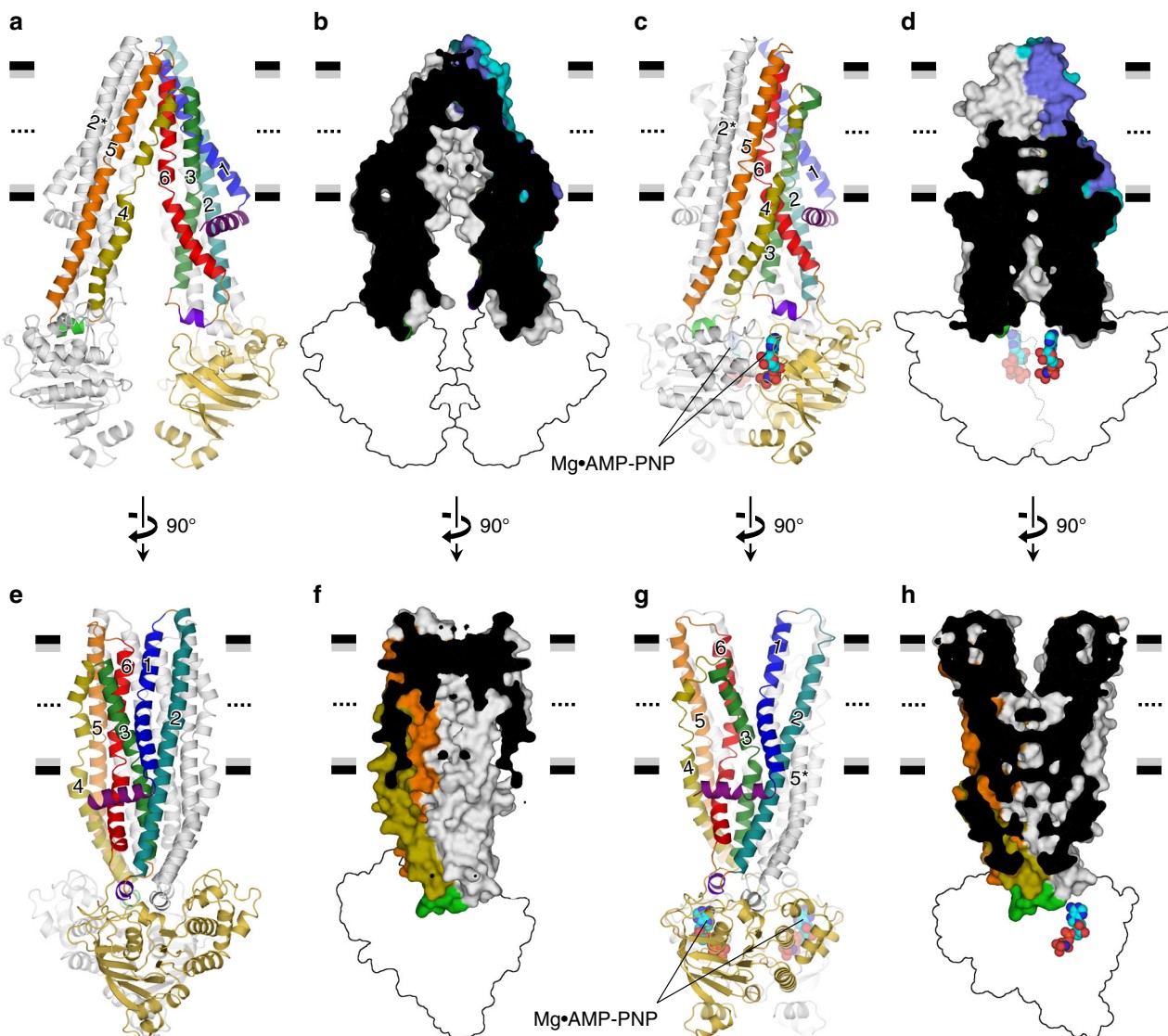

**Fig. 1** Apo inward-facing and Mg$^{2+}$•AMP-PNP-bound outward-facing structures of QTA CmABCB1. Anterior (**a**–**d**) and lateral (**e**, **f**) views of the QTA CmABCB1 structures in the inward- (**a**, **b**, **e**, **f**) and outward-facing (**c**, **d**, **g**, **h**) conformations are shown as cartoon representations (**a**, **c**, **e**, **g**) or as a cutaway surface representations of the TMDs with the interior shown in black (**b**, **d**, **f**, **h**). One subunit is colored, and the other is shown in gray. Secondary structure elements and amino acid residues belonging to the other subunit are indicated by asterisks in all figures. Horizontal black and gray bars represent the expected positions of the hydrophilic and hydrophobic surfaces of the lipid membrane, respectively. In **b**, **f**, **d**, **h**, the cross-section passes through a crystallographic twofold axis. NBDs are shown as outlines for simplicity. In **c**, **d**, **g**, **h**, bound Mg$^{2+}$•AMP-PNP molecules at NBDs are shown as spheres

structural changes forming a substrate pathway from the interior side to the extracellular side. The TM1–TM6 (TM1*–TM6*) intra-subunit interactions mediated by Phe384$^{TM6}$, Phe138$^{TM1}$, and Phe142$^{TM1}$, which close the outlet pathway formed in the center of the bundle, are dissociated in the outward-facing structure by horizontal rotation of the side-chain of Phe384$^{TM6}$ accompanied by twisting of TM6 or TM6* (Fig. 2), which could be induced by tilting of TM5 or TM5*. Concomitantly, the inter-subunit TM6−TM1* or TM6*−TM1 interactions form via the same side chains (Fig. 2, upper panels). Tyr358 in TM5 is also involved in extracellular gating. In the figures, secondary structure elements and amino acid residues belonging to the other subunit are indicated by asterisks.

Comparison of the inside view of the structures in Fig. 3 reveals that close inter-subunit contacts via each pair of side chains (Gln229$^{TM3}$, Tyr233$^{TM3}$, and Lys237$^{TM3}$ in TM3−TM3* (Fig. 3b) and Ala240$^{TM3}$−Gln398*$^{TM6}$ in TM3−TM6* or Ala240*$^{TM3}$ −Gln398$^{TM6}$ in TM3*−TM6 (Fig. 3d)) are formed on the

cytosolic side in the outward-facing structure, leading to the largest change in chamber volume. Because these residues seem to regulate the chamber space, we refer to them collectively as the chamber contraction regulator. The van der Waals contacts of Gln398$^{TM6}$−Ala240*$^{TM3}$ are assisted by those of Gln398$^{TM6}$ −Phe347$^{TM5}$ and hydrogen bonds between Gln398$^{TM6}$ −Arg181*$^{TM2}$ and Arg181*$^{TM2}$−Glu131*$^{TM1}$ (Fig. 3d, Supplementary Fig. 2b). Mutagenesis of Tyr233$^{TM3}$, Lys237$^{TM3}$, A240$^{TM3}$, or Gln398$^{TM6}$ significantly decreased transport activity (Fig. 3e), indicating that those residues participate in the transport function.

The chamber contraction regulator, especially the van der Waals contacts of Gln398$^{TM6}$−Ala240*$^{TM3}$, appears to cause a structural change with spring-like movement of the partially unfolded TM helices, typically TM6 (Fig. 3c, d). In the outward-facing structure, TM6 is deformed and unwound at Gly389$^{TM6}$, Gly392$^{TM6}$, and Gly394$^{TM6}$. We propose that this deformation, a spring-like movement, generates tension to promote the return to

**Table 1 Data collection and refinement statistics**

|  | QTA outward | Hg derivative (QTA outward) | QTA inward |
|---|---|---|---|
| *Data collection* | | | |
| Space group | $P4_132$ | $P4_132$ | $R32$ |
| Cell dimensions | | | |
| $a, b, c$ (Å) | 174.3, 174.3, 174.3 | 176.0, 176.0, 176.0 | 179.7, 179.7, 157.7 |
| $\alpha, \beta, \gamma$ (°) | 90, 90, 90 | 90, 90, 90 | 90, 90, 120 |
| Wavelength | 1.0000 | 1.00789 | 1.0000 |
| Resolution (Å) | 43.6−1.89 (2.00−1.89)[a] | 50−2.70 (2.75−2.70)[a] | 47.1−3.02 (3.20−3.02)[a] |
| $R_{sym}$ | 0.152 (1.187) | 0.148 (0.382) | 0.102 (0.629) |
| Total reflections | 1,517,693 | 1,074,412 | 192,073 |
| Unique reflections | 72,007 | 49,994 | 19,238 |
| $I/\sigma I$ | 13.0 (1.9) | 51.6 (13.1) | 14.3 (2.4) |
| Completeness (%) | 99.2 (95.4) | 100.0 (100.0) | 99.6 (98.5) |
| Redundancy | 21.1 (18.2) | 21.5 (19.7) | 9.98 (10.1) |
| $CC_{1/2}$ (%) | 99.8 (74.0) | (98.3)[b] | 99.9 (91.8) |
| Processing programs | XDS (ver. Jan. 26, 2018) | HKL2000 (ver. 0.98) | XDS (ver. Nov. 3, 2014) |
| *Refinement* | | | |
| Resolution (Å) | 43.6−1.90 (1.95−1.90) | | 47.1−3.02 (3.10−3.02) |
| No. reflections | 67,507 | | 18,276 |
| $R_{work}/R_{free}$ | 0.165/0.208 (0.289/0.293) | | 0.226/0.275 (0.447/0.420) |
| No. atoms | | | |
| Protein | 4537 | | 4476 |
| AMP-PNP | 31 | | 0 |
| $Mg^{2+}$ | 1 | | 0 |
| Detergent | 67 | | 63 |
| Water | 435 | | 13 |
| *B*-factors (Å$^2$) | | | |
| Protein | 62.7 | | 91.3 |
| AMP-PNP | 23.5 | | |
| $Mg^{2+}$ | 22.8 | | |
| Detergent | 68.6 | | 120.2 |
| Water | 46.0 | | 76.4 |
| R.m.s. deviations | | | |
| Bond lengths (Å) | 0.011 | | 0.010 |
| Bond angles (°) | 1.537 | | 1.490 |

The number of crystals for each structure is one
[a]Values in parentheses are for highest-resolution shell
[b]The overall CC1/2 value of the Hg derivative has not been calculated by HKL2000

the inward-facing structure. Several conserved Gly residues in TM6 and TM12 of human P-gp have been proposed to cause helical movement, thereby contributing to transport function[17] and structural flexibility[18].

**TM joints differentiate P-gps from bacterial homologs**. The spatial helical arrangements of the TMDs and the inner chamber shapes of outward-facing conformations differ substantially between CmABCB1 and the bacterial homologs Sav1866[11] and MsbA[12] (Fig. 4 and Supplementary Fig. 2). CmABCB1 has characteristic inter-helical interactions called a "TM1−3 joint", in which TM1 forms close contacts (~4.0 Å) with TM3 via Gly132[TM1]−Ala246[TM3] and Ser128[TM1]−Gly239[TM3], and a "TM3−6 joint", in which TM3 interacts with TM6 (~4.0 Å distance) via Gly251[TM3] sandwiched by Ala386[TM6] and Gly389[TM6]. These small side-chain residues provide tight and restricted contacts between those helices that are maintained in both inward- and outward-facing conformations and function as hinge joints during conformational changes (Fig. 4a). The tight joints among TM1−TM3−TM6 maintain the close inter-subunit distance (9.0 Å) between TM3−TM6* or TM3*−TM6 in the outward-facing state; consequently, inter-subunit van der Waals contacts are formed between Ala240[TM3] and Gln398*[TM6] or Ala240*[TM3] and Gln398[TM6] (Fig. 4c). The human P-gp structure[15] also exhibits close contact between TM1, TM3, and TM6

(Fig. 4d). By contrast, the Sav1866 and MsbA structures have neither TM joints nor close contacts between TM1, TM3, and TM6 at the positions corresponding to the TM joints, and TM1 and TM3 are distant from each other (Fig. 4b, e and Supplementary Fig. 2b). Thus, the equivalent inter-subunit distance between TM3−TM6* or TM3*−TM6 is greater than the side-chain interaction distance (17 Å) between Asn141[TM3] and Ala299*[TM6] or Asn141*[TM3] and Ala299[TM6], corresponding to Ala240[TM3] and Gln398*[TM6] or Ala240*[TM3] and Gln398[TM6] in Sav1866 (Fig. 4e).

The residues involved in TM joints are conserved in eukaryotic P-gps but not in Sav1866 or MsbA (Fig. 4f and Supplementary Fig. 3). When we introduced mutations at key residues of the TM1−3 joint, Gly132[TM1] and Ala246[TM3], the drug resistance IC$_{50}$ of all the mutants drastically decreased (Fig. 4g). Gly191[TM3] and Gly341[TM6] of human P-gp, corresponding respectively to Gly239[TM3] and Gly389[TM6] of CmABCB1, are important for transport functions[19,20]. Accordingly, we propose that these TM joints play crucial roles in structural change of the chamber in the substrate transport cycle of P-gp, suggesting an intrinsic mechanistic difference between multidrug transporters and floppases.

**Conformational change induced upon $Mg^{2+}$-nucleotide binding**. A comparison of outward- and inward-facing states revealed

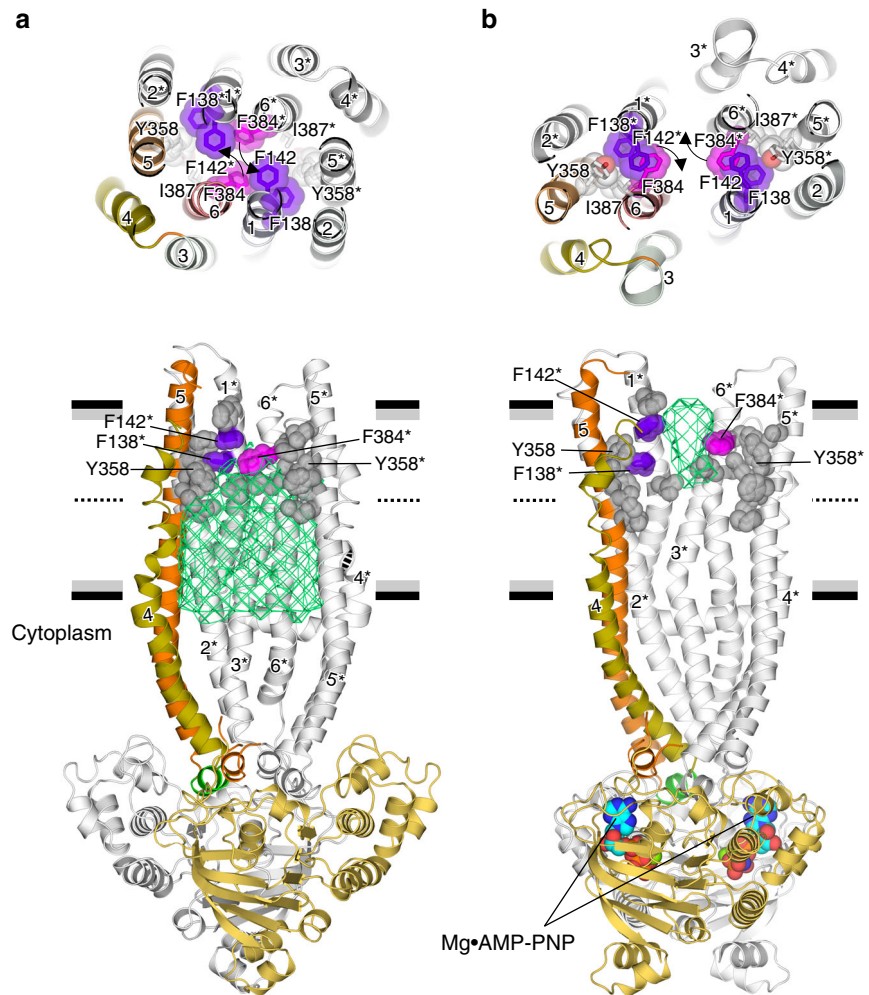

**Fig. 2** Structural changes forming a substrate pathway from the inner side to the extracellular side. Rearrangement of side chains, primarily in TM1, TM6, TM1*, and TM6*, between inward- (**a**) and outward-facing (**b**) conformations of QTA CmABCB1. (Upper panels) Top views of the extracellular gate, showing transition of residues lining TM1, TM6, TM1*, and TM6*. (Lower panels) Lateral views of both structures. For clarity, several TM helices (TM1, TM2, TM3, and TM6) are omitted. One subunit is colored in orange, and the other is shown in gray. The inner chamber in the inward-facing conformation is shown as a green mesh (**a**). In **b**, bound $Mg^{2+}$•AMP-PNP molecules at NBDs are shown as spheres. Horizontal black and gray bars represent the expected positions of the hydrophilic and hydrophobic surfaces of the lipid membrane, respectively

characteristic conformational differences in the NBD and its interface with the TMD, induced by NBD dimerization upon $Mg^{2+}$-nucleotide binding (Supplementary Fig. 4a), which had not been previously reported for B-subfamily ABC proteins. These differences are distributed from the $Mg^{2+}$-nucleotide site in the RecA-like subdomain of the NBD, through the Q-loop and connecting α3 helix in the helical subdomain, up to IH1 and IH2 of the TMD (Fig. 5a and Supplementary Movie 2). $Mg^{2+}$ is bound in optimal octahedral coordination with 2.0−2.1 Å bonds formed by two oxygens from the β- and γ-phosphates of AMP-PNP, the hydroxyl group of Ser485$^{P-loop}$, the side-chain carbonyl group of Gln529$^{Q-loop}$, and two oxygens of water molecules (Fig. 5b and Supplementary Fig. 4c). By contrast, when the NBDs of ABCB2 were crystallized without TMDs[21], the $Mg^{2+}$ coordination position for Gln529$^{Q-loop}$ in CmABCB1 was occupied by a water molecule (Supplementary Fig. 4d), suggesting that the association with the TMD may restrict the movement of the NBD. Because the Q-loop and α3 interact tightly with IH2* via the van der Waals contacts between [Val532$^{Q-loop}$, Phe534$^{Q-loop}$, Tyr544$^{α3}$] and [Thr311*$^{IH2}$, Phe315*$^{IH2}$], and IH2* interacts with IH1 via Ile309*$^{IH2}$ and Phe212$^{IH1}$ in both the inward- and outward-

facing conformational states, $Mg^{2+}$ and Q-loop coordination is likely to trigger the movement of the TMDs through a relay of conformational changes of the Q-loop, α3 helix, IH2*, and IH1 during NBD dimerization (Fig. 5c; Supplementary Movie 2). Furthermore, the adenine moiety of bound AMP-PNP supports the interaction with IH2* and IH1 through hydrogen-bond networks (Supplementary Fig. 4b). Consequently, the structural changes of the NBD upon $Mg^{2+}$-ATP binding are transmitted to the TMD through a relay of van der Waals interactions involving the side chains around IH2* (Fig. 5c). A cryo-EM study of human P-gp also suggested that the Q-loop coordinates $Mg^{2+}$ with the γ-phosphate of ATP and forms part of the interface between the NBD and TMD, interacting with the coupling helices[15].

We also identified a pairing between Glu620$^{α6}$ and Arg644*$^{α7}$ or Glu620*$^{α6}$ and Arg644$^{α7}$ at the bottom of the NBDs, which allows the formation of a salt bridge and stabilizes NBD dimer formation (Fig. 5d). Interestingly, the salt bridges are at the NBD dimer interface near Glu610$^{Walker\ B}$ and His643$^{H-loop}$. The pairing motif, which we term RE-latch, is conserved among the B-subfamily ABC transporters (Supplementary Fig. 3), but not in other subfamilies[22]. The functional importance of RE-latch was

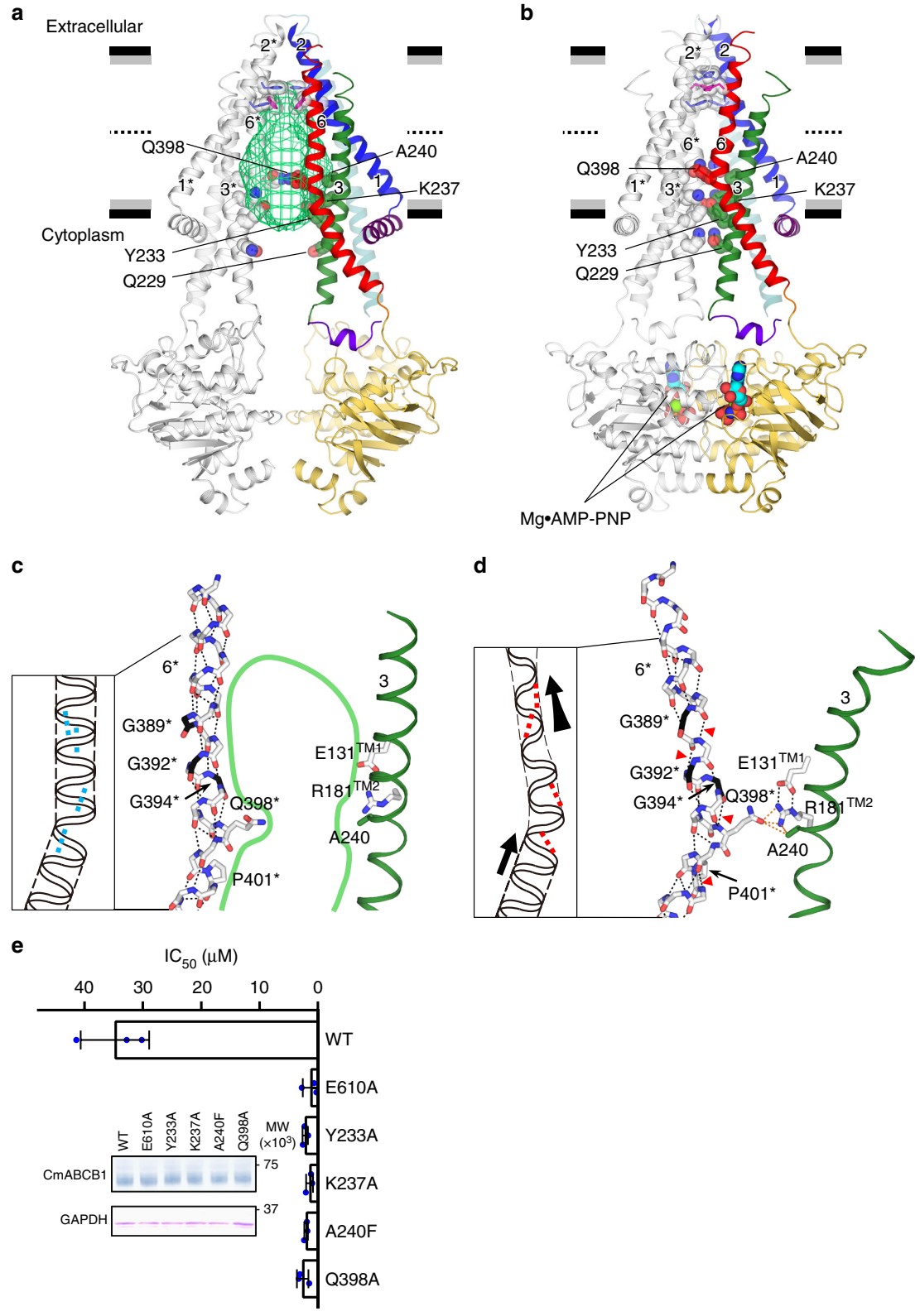

demonstrated by mutagenesis experiments showing that Ala substitution of Glu620 or Arg644 decreased the substrate transport and ATPase activities of CmABCB1 (Supplementary Fig. 5a–c), as well as the ATPase activity of human P-gp (Supplementary Fig. 5d,e). It remains possible, however, that the mutations indirectly affected these activities by altering the

structures around the catalytic residues, Glu610$^{Walker~B}$ or His643$^{H-motif}$, which are located near Glu620$^{\alpha6}$ or Arg644$^{\alpha7}$, respectively (Fig. 5d). The alignment reveals that the residues participating in the Mg$^{2+}$-nucleotide−NBD and NBD−TMD interactions are conserved in transporters whose structures have been elucidated (Supplementary Fig. 3).

**Fig. 3** Large change in the chamber volume. **a**, **b** The inside anterior views of inward- (**a**) and outward-facing (**b**) conformations of QTA CmABCB1. For clarity, TM4, TM5, TM4*, and TM5* shown in Fig. 1a, c are omitted here. Residues forming close inter-subunit contacts on the cytosolic side in the outward-facing conformation are shown as spheres. The inner chamber in the inward-facing conformation is shown as a green mesh (**a**). One subunit is shown in color, and the other is shown in gray. Horizontal black and gray bars represent the expected positions of the hydrophilic and hydrophobic surfaces of the lipid membrane, respectively. In **b**, bound $Mg^{2+} \cdot AMP$-PNP molecules at NBDs are shown as spheres. **c**, **d** Close-up of TM6* and TM3 in inward- (**a**) and outward-facing (**b**) conformations viewed parallel to the membrane. Intra-helical interactions within the main chains of TM6 are shown as dashed lines. In **c**, the internal large cavity facing the intracellular side in the inward-facing conformation is outlined in green. In **d**, TM3 and the side chains of residues interacting with Gln398 are shown as ribbon and sticks, respectively. Polar and van der Waals interactions are shown as black and orange dashed lines, respectively. Non-α-helical hydrogen bonds in TM6 are indicated by red arrowheads. The stretching directions of TM6 from the inward- to outward-facing conformations are shown as thick black arrows in the schematic. **e** Bar graph, overlaid with the actual data points, shows $IC_{50}$ for growth inhibition in a rhodamine 6G susceptibility assay using *S. cerevisiae* AD1-8u⁻ cells. Error bars indicate standard deviation ($n = 3$). Cells expressing the ATPase-deficient mutant E610A served as controls. Inset shows the amounts of mutant and WT CmABCB1 expressed in AD1-8u⁻ cells, as determined by western blotting. Uncropped images of the blots are shown in Supplementary Fig. 6

## Discussion

The conformational change between the apo inward-facing and $Mg^{2+}$-nucleotide-bound outward-facing X-ray structures provides insight into the mechanism of multidrug transport. We propose that the active transport of diverse substrates by P-glycoprotein is mediated by alternative contraction and dilation of the inner chamber (Fig. 6). These changes are promoted by the dimerization and dissociation cycles of the two NBDs, coupled with $Mg^{2+}$-ATP binding and hydrolysis. From the inward-facing to the outward-facing conformation, $Mg^{2+}$-ATP binding to the NBDs causes a conformational change in the TMDs mediated by NBD−TMD interactions, aided by a relay of van der Waals interactions among the side chains of residues in the helical subdomain of the NBD, coupling helices, and TM helices upon Q-loop−$Mg^{2+}$-ATP interaction. Consequently, the volume of the chamber is significantly decreased by TM3−TM3*, and both the TM3−TM6* and TM3*−TM6 interactions are manipulated by the chamber contraction regulator (Gln398 and Ala240) and TM joints (TM1−TM3 and TM3−TM6). The conformational change simultaneously breaks the TM1−TM6 interaction, consisting of the van der Waals contacts and hydrogen-bonding network in the cluster of aromatic hydrophobic side-chains[9], and opens the extracellular gate via twisting of TM6 and TM6*. It is possible that the opening of the extracellular gate liberates the substrate to the extracellular space, and that the RE-latch (Glu620 and Arg644) stabilizes the outward-facing conformation until the substrate is expelled. Because the potential energy generated by NBD dimerization upon ATP binding is stored in the structural strain of the TMDs, especially TM3, TM3*, TM6, and TM6*, when ATP is hydrolyzed, the potential energy must be transformed into force to return from the outward-facing to the inward-facing conformation. The narrowness of the opening of the extracellular gate in our structure of the outward-facing state suggests that substrates are transported to the extracellular space rather than into the lipid bilayer (Supplementary Fig. 2a).

In general, the outward-facing high-resolution crystal structure of $Mg^{2+}$-nucleotide-bound homodimeric CmABCB1 is similar to the cryo-EM structure of monomeric human P-gp[15], suggesting that common substrate recognition and structural change mechanisms are involved in multidrug export. Despite the molecular constitution of the dimer or monomer, both substrate binding sites have a cluster of hydrophobic aromatic residues that can offer adaptive plasticity and gating through the extracellular space. Residues able to form H-bonds, corresponding to Gln147 and Thr381 of CmABCB1, are found in the TMD1 of mammalian P-gp (Supplementary Fig. 3), implying similar stabilization of the inward-facing conformations. However, the residues participating in chamber regulation differ between human P-gp and CmABCB1. Asymmetric P-gp has a substrate binding site regulated by additional inhibitors and modulators[23,24], and sometimes

two NBDs exhibit asymmetric behavior in ATP binding and hydrolysis[10,14,25]. The structure of chimeric P-gp bound to zosuquidar[26] also revealed some differences from the apo inward-facing CmABCB1 structure: both TM4 and TM10 structures must be structurally flexible, but the observed conformational change triggered by NBD dimerization is different. These phenomena could be due to an intrinsic difference in the molecular architecture or the effect of antibody binding to chimeric P-gp.

In the case of P-gp, inward-facing conformations are more stable than outward-facing conformations with bound ATP, whereas in the bacterial homologs MsbA and Sav1866, the two states have similar conformational stability[13,14]. The difference in the conformational stability of P-gp may permit the existence of short-lived outward-facing conformations during ATP hydrolysis, followed by rapid reversal to the inward-facing state after hydrolysis[14]. Until now, however, the structural and thermodynamic factors underlying the differences in the stability of states between the P-gp and bacterial homologs, despite similarities in their molecular structures, have remained enigmatic.

Our study provides some answers. First, our structures of two states of the same P-gp reveal profound structural differences in the outward-facing states of CmABCB1 and bacterial homologs that function as floppases. Secondly, key structural determinants that are conserved in mammalian P-gps underpin the mechanism by which ATP binding and hydrolysis initiate local structural changes that are propagated throughout the entire molecule, from NBD to TMD. The instability of the outward-facing conformation seems to arise from accumulated elastic strain in the coiled spring-like transmembrane helices (Fig. 3c, d), which is caused by NBD dimerization upon ATP binding. Release of this strain drives a rapid reversal to the inward-facing state upon hydrolysis.

## Methods

**Chemicals**. Substrates for CmABCB1 were purchased from Wako, except for tetraphenylphosphonium bromide (Tokyo Chemical Industry), monensin (Alexys), and calcein AM (Nacalai).

**Construction of CmABCB1 and human P-gp mutants**. CmABCB1 and human P-gp mutants were made by PCR site-directed mutagenesis. Five CmABCB1 mutants, E610A, Y233A, K237A, A240F, and Q398A, were constructed by QuikChange site-directed mutagenesis kit (Stratagene), and the other five mutants, Q147A/T381A, G132V, A246V, E620A, and R644A, were by QuikChange Multi system (Stratagene) using corresponding primers (Supplementary Table 2) and pABC3 harboring *CmABCB1* gene with a C-terminal FLAG and His₆ affinity tags[9]. Human P-gp mutants were made by PrimeSTAR mutagenesis (Takara Bio) and In-Fusion HD cloning (Clontech) using corresponding primers (Supplementary Table 2) and the pcDNA3.1 vector harboring human *P-gp* gene with FLAG tag[27], which was modified by the insertion of the TEV protease cleavage site and the His₁₀ affinity tag at 5′ region of FLAG tag with oligo nucleotides (Supplementary Table 2) and the restriction enzyme (Not I). All mutants were confirmed by DNA sequencing.

**Expression of CmABCB1**. To obtain wild-type and mutants of CmABCB1, the proteins were expressed in *Saccharomyces cerevisiae* AD1-8u⁻ cells[28]. The cells

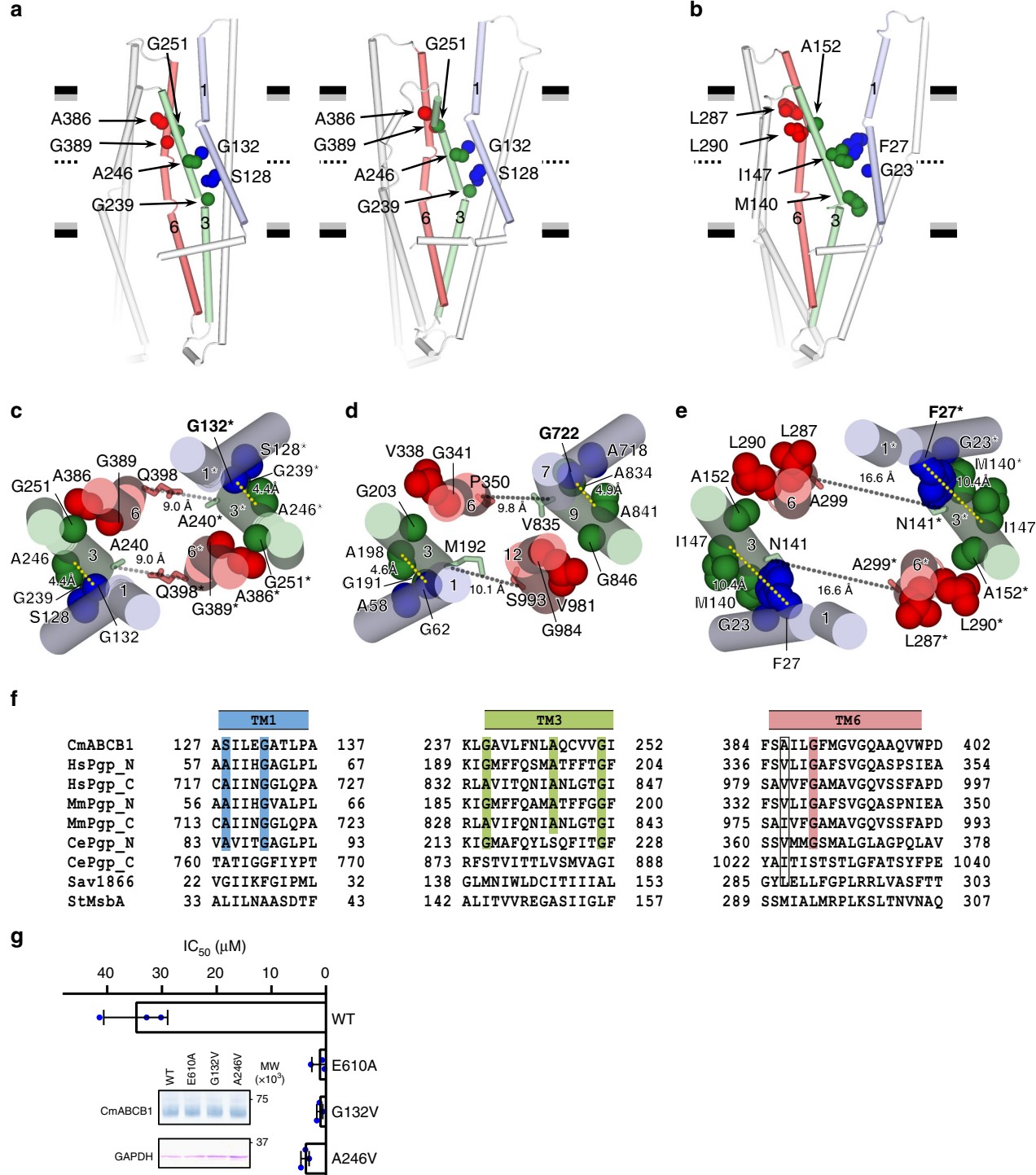

**Fig. 4** TM joints characterizing the outward-facing structure of P-gp. **a**, **b** Inter-helical interactions of TM1−TM3 and TM3−TM6 of inward- (**a**, left) and outward-facing (**a**, right) conformations of QTA CmABCB1, and comparison with outward-facing structure of Sav1866 (**b**). Residues serving as TM joints in CmABCB1 and corresponding residues in Sav1866 are shown as spheres. Only the TMD of one subunit is shown for simplicity. **c**–**e** Arrangement of TM joints of outward-facing QTA CmABCB1 (**c**) and human P-gp (**d**) viewed from the extracellular side and comparison with Sav1866 (**e**). TM1 (TM1* or TM7), TM3 (TM3* or TM9), and TM6 (TM6* or TM12) are shown as cylinders. Residues serving as TM joints in CmABCB1 and the corresponding residues in human P-gp and Sav1866 are shown as spheres. **f** Local sequence alignment of the TM joints consisting of TM1, TM3, and TM6. Conserved Gly and other residues with small side chains, such as Ala, and Ser, are highlighted. **g** Bar graph, overlaid with the actual data points, shows IC$_{50}$ for growth inhibition in rhodamine 6G susceptibility assay using *S. cerevisiae* AD1-8u− cells. Error bars indicate standard deviation (*n* = 3). Cells expressing the ATPase-deficient mutant E610A served as controls. Inset shows the amounts of mutant and WT CmABCB1 expressed in AD1-8u− cells, as determined and analyzed by western blotting. Uncropped images of the blots are shown in Supplementary Fig. 6

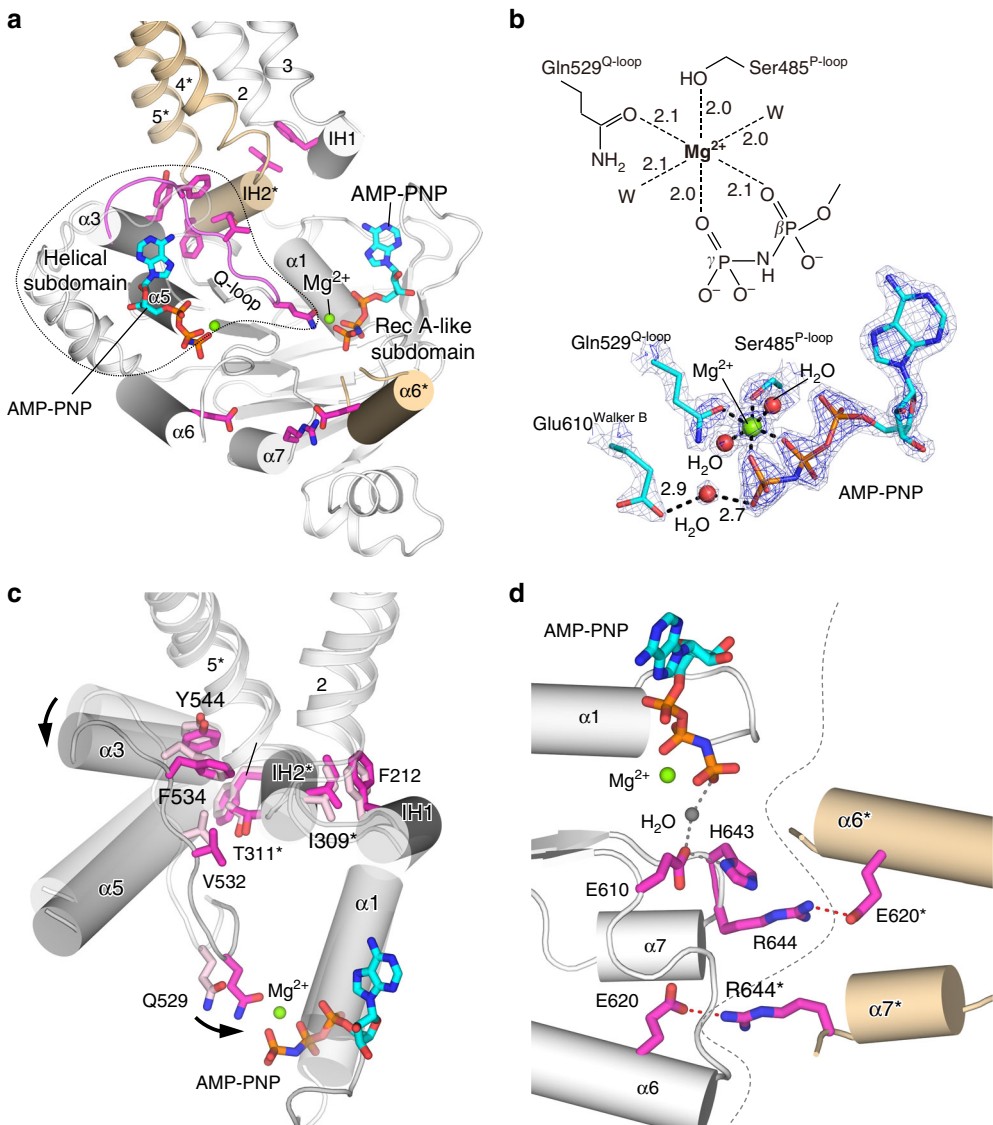

**Fig. 5** Coupling between NBD−TMD and NBD−NBD* in the outward-facing conformation of QTA CmABCB1. **a** NBD bound to $Mg^{2+}$•AMP-PNP in a subunit of QTA CmABCB1, viewed parallel to the membrane. The NBD and TM2−IH1−TM3 of one subunit are shown in white, except for Q-loop-α3, shown in magenta; TM4*−IH2*−TM5* and α6* of the other subunit are shown in orange. The key residues for coupling between NBD−TMD and NBD −NBD* are shown as magenta sticks. **b** Octahedral coordination of $Mg^{2+}$. Light blue and blue meshes represent the 2Fo−Fc map contoured at 2.7 and 4.5 sigma, respectively. Polar interactions are shown as dashed lines. **c** Superposition of the NBDs in the outward- (gray) and inward-facing (white) conformations, based on the RecA-like subdomain. The direction of movement of α3 and Gln529 between the inward- and outward-facing conformations are shown as arrows. Residues contributing to the coupling of movement between the NBD and TMD are shown as sticks. **d** Interaction between NBDs stabilized by RE-latch. The salt bridges in RE-latch are shown as red dashed lines, and the boundary between the two NBDs is shown as gray dashed lines

were precultured in 20 ml of extract-peptone-dextrose (YPD) medium for overnight at 30 °C with shaking at 240 rpm (BioShaker BR-23FP, TAITEC). The precultured cells were inoculated in 400 ml of YPD medium to an $OD_{600}$ of 0.5, and were cultured at 30 °C up to an $OD_{600}$ of 5. The cells were then diluted in 8 L of YPD medium to an $OD_{600}$ of 0.1, and were grown in eight 2.5-L flasks (Thomson) at 25 °C with shaking at 240 rpm using an Innova 4330 incubator shaker (New Brunswick Scientific). After 24 h, cells were harvested by centrifugation (3000 × g, 15 min) and stored at −80 °C until use.

**Purification of CmABCB1.** The yeast cells were thawed on ice and disrupted using an EmulsiFlex-C3 (Avestin) at 25,000 psi in a buffer containing 20 mM Tris-HCl (pH 7.0) and 150 mM NaCl. The homogenate was clarified by centrifugation at 1500 × g for 15 min to remove the unbroken cells and nuclei, and then crude membranes were collected by ultracentrifugation (100,000 × g, 1 h). The membranes were mechanically homogenized and subsequently solubilized for 1 h in a binding buffer (20 mM Tris-HCl (pH 7.0), 300 mM NaCl, 20 mM imidazole) containing 1% (w/v) polyoxyethylene(9)dodecyl ether ($C_{12}E_9$) (Wako). Insoluble

material was removed by ultracentrifugation (100,000 × g, 1 h) and immobilized metal−ion affinity chromatography (IMAC) resin (Bio-Rad) was added to the supernatant. After a 3-h incubation, the bound protein was eluted with binding buffer containing 300 mM imidazole. After cleaving the flexible N-terminal region (1−92) by trypsin treatment[9], the IMAC-purified CmABCB1 were further purified by gel filtration using a Superdex200 column (GE Healthcare) equilibrated in a buffer composed of 20 mM Tris-HCl (pH 7.0), 150 mM NaCl, and 0.2% (w/v) n-decyl-β-D-maltopyranoside (βDM) (Anatrace). Peak fractions were pooled and concentrated to 10 mg ml⁻¹ for crystallization experiments. In the preparation used for ATPase activity assays, membranes (5 mg ml⁻¹ protein) were solubilized using 1% (w/v) n-dodecyl-β-D-maltopyranoside (βDDM) (Anatrace) instead of $C_{12}E_9$ and purified in 0.05% (w/v) βDDM without the trypsin treatment.

**Crystallization.** Crystallization was carried out by the sitting-drop vapor diffusion method at 20 °C. For crystallization of the outward-open form, the QTA mutant was preincubated for 1 h with 10 mM AMP-PNP and 20 mM $MgCl_2$. Crystals of outward-open QTA were grown by mixing protein (10 mg ml⁻¹) with an equal

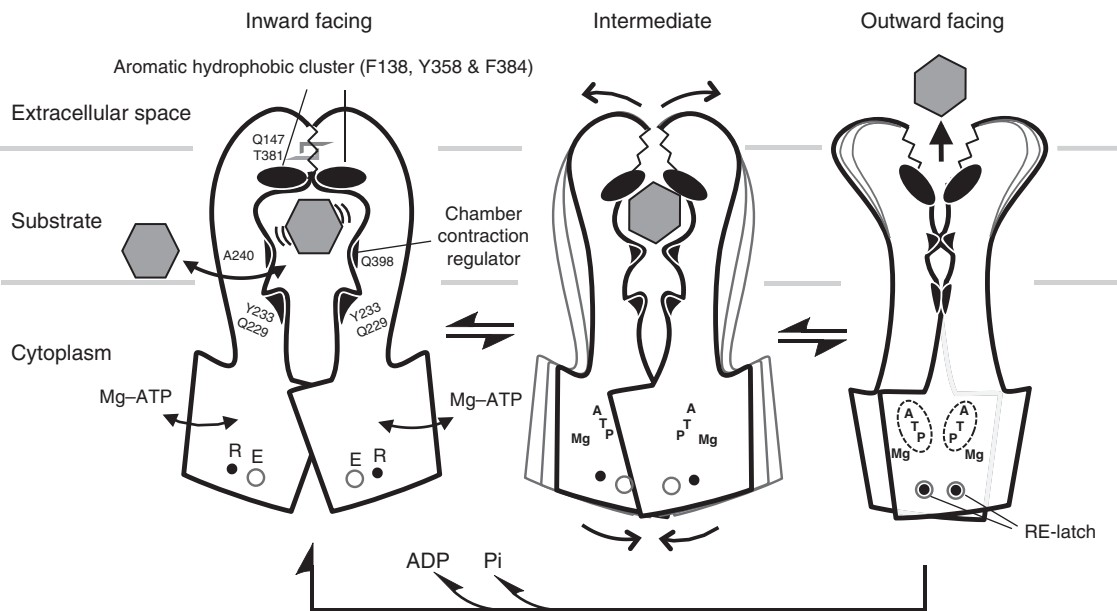

**Fig. 6** Proposed model for transport mechanism of CmABCB1. Schematic drawing of conformational change between the inward- (left) and the outward-facing (right) CmABCB1. An expected intermediate state is depicted in the center panel. The characteristic devices indispensable for the mechanics of transport are represented

volume of reservoir solution containing 19−21% PEG 2000 MME, 50 mM potassium nitrate (pH 7.4), and 50 mM magnesium nitrate (pH 4.1). Crystals were cryoprotected by increasing the PEG 2000 MME concentration to 30%, followed by increasing the 1,4-butanediol concentration to 5%. Cryoprotected crystals were flash-frozen and stored in liquid nitrogen. Mercury-derivative crystals of outward-open QTA were prepared by soaking the native crystals in reservoir solution supplemented with 1 mM mercury chloride for 1 day, followed by back-soaking. Crystals of inward-open QTA were grown by mixing protein (10 mg ml$^{-1}$) with an equal volume of reservoir solution containing 14% PEG 2000 MME and 100 mM magnesium nitrate (pH 4.1). Cryoprotection was performed as described above.

**Data collection and structure determination**. X-ray diffraction data sets of outward-open, Hg derivative, and inward-open QTA crystals were collected using an MX225HE detector (for outward-open QTA: camera length, 200 mm; scan step, 0.5°; exposure time, 0.5 s; scan angle, 90°; attenuator, Al 600 μm; for Hg derivative: camera length, 260 mm; scan step, 1.0°; exposure time, 1.0 s; scan angle, 180°; attenuator, Al 400 μm) or a PILATUS 6M detector (for inward-open QTA: camera length, 600 mm; scan step, 0.5°; exposure time, 0.5 s; scan angle, 180°; attenuator, Al 1140 μm), at beamline BL41XU of SPring-8 (Hyogo, Japan). Diffraction data for inward-open and outward-open QTA were processed using XDS[29], and data for the Hg derivative were processed using HKL2000 [30]. The initial phases for the outward-open and inward-open structures of QTA were solved by single-wavelength anomalous diffraction using a mercury-derivative and molecular replacement, respectively. Each final model was obtained by an iterative process of manual model-building with COOT[31] and refinement with REFMAC5[32] and PHENIX[33] against X-ray diffraction data. Details of data collection and refinement statistics are summarized in Table 1. Molecular graphics were rendered in PyMOL[34].

**Drug susceptibility assay in yeast cells**. Transport activity of CmABCB1 was assayed by assessing the drug susceptibility of *S. cerevisiae* AD1-8u⁻ cells expressing wild-type or mutant CmABCB1[9]. The cells were precultured in YPD medium at 30 °C for 16 h. The precultured cells were inoculated in YPD medium to an OD$_{600}$ of 0.5. The cells were then cultured at 30 °C up to an OD$_{600}$ of 2−3. The cultured cells were diluted in YPD medium to an OD$_{600}$ of 0.2, and 50 μl of each cell suspension was inoculated into 450 μl YPD medium containing a drug at the indicated concentration in a 96-well V-shaped MasterBlock (Greiner Bio One). After culture at 30 °C for 13−14 h in YPD medium containing rhodamine 6G, etoposide, itraconazole, tetraphenylphosphonium, monensin, or fluconazole, optical densities of the cell suspensions were measured at 600 nm. The assay was performed using 4−8 clones, and the averages and standard deviations of measured values were calculated. Expression levels of mutant proteins were evaluated by western blotting analysis with anti-His (1014992, Qiagen, 1:5000) or anti-GAPDH antibody (MA5-15738, Invitrogen, 1:5000). Cells expressing ATPase-deficient mutant E610A were used as negative controls.

**Expression and purification of human P-gp**. WT and mutant P-gps were transiently expressed by PEI-mediated transfection[35] in suspension culture-adapted HEK293 cells (FreeStyle 293-F cells; Thermo Fisher). Cells were transfected with a mixture of PEI-MAX (Mw 40,000; Polysciences) and plasmids at final concentrations of 4 μg ml$^{-1}$ and 1 μg ml$^{-1}$, respectively. For purification, P-gp-expressing cells were solubilized with 1% C$_{12}$E$_8$ in solubilization buffer (50 mM 4-(2-hydroxyethyl)-1-piperazineethanesulfonic acid (HEPES)-Na (pH 7.2), 150 mM NaCl, 50 mM KCl, 10% glycerol, and 1 mM 2-mercaptoethanol) supplemented with protease inhibitors (Complete EDTA-free; Roche Applied Science). After insoluble materials were removed by centrifugation (45,000 × g, 30 min), ANTI-FLAG M2 Affinity Gel (A2220, Sigma-Aldrich) was added, and the sample was incubated for 2 h. Proteins were eluted in solubilization buffer containing 0.05% C$_{12}$E$_8$ and 0.15 mg ml$^{-1}$ each of FLAG peptide and 3× FLAG peptide, and then concentrated to 0.5−2 mg ml$^{-1}$ using Amicon Ultra 0.5 ml filters (100,000, Merck Millipore).

**ATPase measurements**. ATPase activity of purified CmABCB1 in βDDM micelles was measured in 50 mM Tris-HCl (pH 7.5), 150 mM NaCl, 0.05% βDDM, and 10 mM MgCl$_2$ with or without ATP, containing the indicated compounds. The ATPase activity of CmABCB1 reconstituted in proteoliposomes was measured under the same conditions, except that βDDM was omitted. To prepare liposomes, egg L-α-phosphatidylcholine lipids (Avanti) dissolved in chloroform were dried and hydrated with buffer containing 20 mM Tris-HCl (pH 7.5), 150 mM NaCl, 5 mM MgCl$_2$, and 2 mM dithiothreitol. The hydrated lipid suspension was subjected to five freeze-thaw cycles and sonicated in a bath sonicator until the suspension clarified. To reconstitute CmABCB1 into liposomes, 1.0 mg ml$^{-1}$ purified CmABCB1 in buffer containing 20 mM Tris-HCl (pH 7.5), 150 mM NaCl, and 0.05% βDDM was mixed with an equal volume of 10 mg ml$^{-1}$ lipid suspension, diluted to 50 μg ml$^{-1}$ protein concentration in buffer containing 20 mM Tris-HCl (pH 7.5) and 150 mM NaCl, and then incubated at 23 °C for 20 min. The ATP hydrolysis reaction of CmABCB1 was performed at 37 °C, and the initial hydrolysis rate was measured by detecting inorganic phosphate released from ATP using a colorimetric method[9]. The ATPase activities of purified CmABCB1 determined in detergent micelles were similar to those in liposomes (Supplementary Fig. 1d,e) and those of purified human P-gp determined in liposomes[36].

To analyze the ATPase activity of human P-gp, purified proteins were reconstituted in egg lecithin liposomes, and ATPase reactions were carried at 37 °C for 30 min in reaction buffer (40 mM Tris-Cl (pH 7.4), 100 mM NaCl, 0.1 mM ethylene glycol-bis(2-aminoethylether)-N,N,N',N'-tetraacetic acid (EGTA), 2 mM dithiothreitol, 3 mM MgCl$_2$, and 3 mM ATP). Reactions were stopped by the addition of EDTA, and released adenosine diphosphate (ADP) was measured by high performance liquid chromatography (HPLC)[37].

**Determination of kinetic parameters for the ATPase reaction**. Three different analyses were performed using the following equations[9]. When the ATPase assay was performed using various concentrations of ATP, kinetic parameters were

determined using the Michaelis−Menten Eq. (1):

$$\nu = \frac{k_{\text{basal}}[e][s]}{K_{\text{m}}^{\text{ATP}} + [s]}. \tag{1}$$

The ATPase assay was performed in the presence of various concentrations of substrates (rhodamine 6G or verapamil) and 5 mM ATP. Kinetic parameters were determined using Eq. (2) for WT or RE-latch mutants of CmABCB1 or human P-gp, and using Eq. (3) for G132V, A240F, A246V, or Q398A mutants of CmABCB1:

$$\nu = [e]\left(k_{\text{basal}} + \frac{(k_{\text{sub}} - k_{\text{basal}})[s]}{K_{\text{m}}^{\text{Drug}} + [s]}\right)\left(1 - \frac{[s]}{K_{\text{i}}^{\text{Drug}} + [s]}\right), \tag{2}$$

$$\nu = [e]\left(\frac{(k_{\text{basal}} - k_{\text{sub}})}{1 + \frac{[s]}{K_{\text{i}}^{\text{Drug}}}} + k_{\text{sub}}\right). \tag{3}$$

Here, $\nu$ is the initial ATP hydrolysis rate; $[e]$ is the concentration of CmABCB1 or human P-gp; $[s]$ is the concentration of ATP or substrates; $k_{\text{basal}}$ and $k_{\text{sub}}$ are the catalytic rate constants of ATPase activity in the absence and presence of substrate, respectively; $K_{\text{m}}^{\text{ATP}}$ is the Michaelis constant for ATP, $K_{\text{m}}^{\text{Drug}}$ is the apparent Michaelis constant for substrate activation, and $K_{\text{i}}^{\text{Drug}}$ is the apparent inhibition constant for substrate inhibition. Fitting was carried out using GRAFIT (Erithacus Software) or KaleidaGraph (Synergy Software).

**Reporting summary**. Further information on experimental design is available in the Nature Research Reporting Summary linked to this Article.

## Data availability
Data supporting the findings of this manuscript are available from the corresponding author upon reasonable request. Atomic coordinates and structural factors of the QTA CmABCB1 in the AMP-PNP-Mg$^{2+}$-bound outward-facing state and in the inward-facing apo state have been deposited in the Protein Data Bank under accession codes 6A6M and 6A6N, respectively.

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

## Acknowledgements
The X-ray experiments were performed at BL41XU and BL32XU of SPring-8 with the approval of the Japan Synchrotron Radiation Research Institute, JASRI (Proposal Nos. 2013B1277, 2014A1163, 2014B1001, 2014B1772, 2015A1039, 2015B2039). We thank the beamline scientists at SPring-8, especially T. Kumasaka, K. Hasegawa, and H. Okumura, for assistance with data collection; R. Hirokane for assistance with diffraction data processing; A. Fujioka for assistance with the yeast drug susceptibility assay; and S. Masuko for assistance with DNA construction. This work was supported by JSPS KAKENHI Grant Numbers 25251006, 26102724, 26670014, and 17H03664 (H.K.); 25221203 and 18H05269 (K.U.); 24689029 and 17K07306 (A.K.); 26840048 and 16K07320 (T.Y.); and 15H01638 (Y.K.); and by the X-ray Free-Electron Laser Priority Strategy Program (MEXT) (H.K.).

## Author contributions
H.K. and K.U. planned the project. H.K. supervised the study, and carried out the structural analysis. A.K. performed DNA manipulations and carried out protein expression, purification, and crystallization. T.N. and A.K. performed X-ray data analysis, model-building, and refinements. T.Y. performed biochemical experiments and

structural analyses. K.M. performed mutagenesis and functional experiments with the TM joints. Y.K. performed experiments with human P-gp. T.Y., A.K., and H.K. wrote the paper on the basis of discussions with T.N., Y.K., and K.U. All authors approved the final manuscript.

## Additional information

**Competing interests:** The authors declare no competing interests.

