## [Peer Review File · Nature Communications]

Reviewers' Comments:

Reviewer #1:

Remarks to the Author:

The changes have improved the manuscript dramatically. The figures and movies are also much improved and in general very clear, especially considering the complexity of some of the items being illustrated.

I have remaining concerns with the descriptions and interpretations. Two fundamental issues are likely at the root of these problems. First, the authors still try to describe their model using macroscopic analogies such as "squeezing" and "contraction" that are inappropriate at the molecular scale: matter is essentially incompressible at the molecular scale. Second, the two new structures are both "apo" structures with respect to the transported substrate. Thus, it is likely that the differences between the two structures more accurately suggest the transition path between the outward-facing state after substrate release to the inward-facing state before substrate binding, rather than the transition that the authors seem most interested in – going from inward-facing to outward-facing in the presence of substrate followed by substrate release. I believe Reviewer 1 had similar concerns when stating "In addition, I am not convinced by the model of extrusion presented. There may exist multiple OF intermediates and without a mechanistic context one can imagine a multitude of mechanisms." In other words, the transition from inward- to outward-facing could well include outward-facing states with large substrate-containing cavities. These might be less stable in the absence of substrate (after the substrate unbinds and diffuses outward), leading to a high probability of transitioning to the observed crystalline state with a contracted chamber. Such an order of events makes more sense in microscopic terms, effectively generating a Brownian ratchet that insures substrate is released and the transport cycle continues.

To rephrase this last point: recurring in the manuscript is the idea of "squeezing/extruding/pumping" that seems to be used in the macroscopic sense. If this is taken literally, it is then reasonable to ask whether a measurement of a physical force on the substrate can be made as it is extruded. In contrast, with microscopic reversibility and a Brownian ratchet there is no directed force on the molecule, but the equilibrium corresponding to the substrate binding and unbinding in some outward-facing state leads to a dissociation of the substrate and an outward-facing state that relaxes into a low-affinity apo state (which likely corresponds to the one observed in the crystal structure).

Specific comments:

1. The revised title is still inappropriate, considering the comments above.
2. Lines 10-14 of the abstract should be revised considering the comments above.
3. P. 4 line 16: "two types of inner gates" is rather confusing; "inner" is often used to refer to intracellular side of the membrane, for example, but here it seems to mean "internal"? Perhaps simplifying to "two gates" makes more sense?
4. P. 4 line 9: ref(s) needed about side chains involved in substrate and inhibitor binding.
5. p. 5 lines 8-9: It's unclear to me what the authors mean by "which reaches beyond the intracellular membrane".
6. p. 8 lines 10-11: "unwound and relaxed" and "regular and rigid" – the first qualifier is structure-based and makes sense. But the second one ("relaxed" and "rigid") requires additional evidence. An unwound region is not necessarily "relaxed", and "rigid" similarly is not warranted, unless there is evidence such as high or low B-factors in comparison to other elements in the structure, etc. I suggest simplifying the sentence by simply using "unwound" and "regular".
7. p. 9 lines 12-14: The description includes a time or order-of-events component that hasn't been demonstrated (twisting of TM6 "induced by" tilting of TM5; "Simultaneously"). The structures make it clear that all these structural changes do happen to reach the observed conformation, but the order in which they happen cannot be inferred simply from looking at the structure.
8. p. 12 lines 11-12: "tension generated by the spring-like movement"? This is again a prediction that is disguised as an inference. It could be ok to edit to "and we propose that this deformation, a spring-like movement, generates tension to promote...".
9. p. 13 line 8: this sentence seems supported only by Supp Fig. 4B, not 4A.

10. p. 14 lines 7-9: I appreciate that the authors focus on analyzing sequences of transporters for which there are available structures to making inferences about specific structural features and the nearby sequence features, such as the contacts observed in TM1, TM3, TM6. However, to state that a given residue is conserved “across the B-subfamily ABC transporters” based on only this very limited and biased alignment is inappropriate (although it may be accurate, the evidence presented in Supp Fig. 3 is not enough to support it). The authors should either rephrase (these contacts are conserved in the transporters whose structures have been elucidated, which is the scope of the alignment) or use an alignment that more completely and deeply samples the B-subfamily transporters.

11. p. 14-15: the description of the proposed mechanism should be edited to address the above comments about macroscopic analogies vs. microscopic reversibility, and about which conformational transitions are most likely suggested by comparison of their two structures. Particularly, but not only, p. 14 lines 14-15, p. 15 lines 12-19. Also, the final sentence of the manuscript, p. 18 lines 3-4.

12. p. 16 lines 6-8: the conformation of the transporter in an outward-facing state with bound substrate (although it may be challenging to observe depending on how stable it is) is needed to make a statement about this. For all we know, there is a transient state with openings to the lipid bilayer.

13. Table 1: The overall CC1/2 value is missing for the Hg derivative.

14. In Supp. Fig. 1 and Supp. Fig. 5, ATPase activity assays are done at (n = 2) and SD means very little when using only two measurements. In such a situation, it is generally more prudent to report the two measurements as replicates (plotted as two points, instead of bars/points with error bars) without using mean +/- SD.

Reviewer #2:

Remarks to the Author:

The manuscript by Kodan et al. reports two crystal structures of a eukaryotic homolog of Pgp, cmABC B1. One of these structures is referred to as outward-facing which represents the first such structure of a Pgp homolog at high resolution. Starting from their previous inward-facing structure of this protein, the authors designed a mutant (QTA) which stabilize the OF. In addition to describing the structure, the manuscript contains instructive comparisons with bacterial ABC exporters as well as recently determined cryoEM structure of human Pgp. While the latter structure somewhat diminishes the impact of this work, there are atomic details that provide insights into the stability of this state as well as the mechanism of alternating access.

The current version of the manuscript is much improved with the speculative aspects toned down. But there are still concerns that the authors need to address:

1- In both title and abstract, the authors must note that this is a eukaryotic homolog of Pgp. This is important in light that this is a homodimeric ABC transporter.

2- The authors suggest that the double mutation of Q147ATM1/T381ATM6 “had minimal effect on protein function of drug resistance and ATPase hydrolysis activity.” In another sentence they claim “...the QTA mutation only affected the conformational equilibrium between inward-facing and outward-facing states.” This is mutually exclusive. If the energy landscape is modified then there has to be a functional consequence. Indeed in supplementary Figure 1, the results show the QTA mutant has a higher stimulated ATPase activity (more than 30%) in the presence of Rhodamine and a lower maximal ATPase hydrolysis (30% less) compared with the wild-type CmABAB1. This is inconsistent with the authors’ statement. The fact that the mutant may have minimal impact on drug transport in vivo does not suggest unaffected mechanism of function in vitro, as clearly demonstrated here. Rather than shying away from this result, the authors should put it in mechanistic context.

3- A cluster of residue referred to as the “chamber contraction regulator” are proposed to stabilize the OF. First, I do not see a reason for the term particularly that it is speculative. The experiments to test the role of these residues involve mutations to alanine. While the substitutions do affect transport, they are not specific to the mechanism proposed. In fact the drastic changes render

these experiments of limited value. The interpretation should be toned down.

4- On a more fundamental level, what is the basis of referring to the structure as OF? In fact the opening to the outside is limited as noted by the authors. This is strikingly illustrated with the comparison to the two bacterial homodimers. Both this structure and human Pgp are less open and have diffuse density in this area. The manuscript suggests that this may be an intrinsic mechanistic difference between multidrug transporters and floppases. An alternative interpretation is that these structures may reflect a mostly occluded conformations prior to the OF state. In fact the DEER study cited by the authors propose the presence of this intermediate. The authors should discuss this possibility.

5- There are a number of loosely phrased, partially inaccurate statements that require attention

a- In the abstract, the statement that "lack of structural information regarding other conformational states..." is not accurate. There is now a structure of Pgp bound to an inhibitor and one in a putatively OF conformation.

b- abstract needs to be rewritten to make a crisp distinction between mammalian Pgp and the homodimer under study in this manuscript

c- P4 line 12 " no outward facing crystal structure" but there is a cryoEM structure

d- page 4 line 17: the use of the word "gates" imply a specific mechanism, please change.

e- page 6, line 12 "this ensures....hydrolysis". This statement is ambiguous. How does a free energy process ensure a specific rate? How is the return to inward facing conformation following ATP hydrolysis irreversible?

f- Page 13 line 19 : the role of the RE latch is speculative. The sentence starting from RE-latch and ending online 1 page 14 should be deleted. No information on kinetics is available in this manuscript.

g- I find the section on mechanism of transport too long. The author should summarize their findings crisply stating clearly that what they have is a structural mechanism of alternating access and not transport. I find the discussion about how the drug is exported speculative because of the lack of an intermediate state structure that informs on how the substrate threads its way to the outward-facing cavity. MsbA and Pgp have IF and OF structures. The only added insight in this manuscript comes from the high resolution and the ability to discern subtle structural aspect. The authors are encourage to avoid excessive speculations on how substrates are pushed through.

Response to Reviews

We are pleased that the reviewers approved of the improvements in the revised version of the manuscript. We thank the reviewers for their careful evaluation and constructive feedback. As outlined below, we have addressed all of the points they raised. Based on their comments, we changed the title and moderated the tone of our speculative claims. We have attached the revised manuscript, in which the revised portions of the text are marked in red. We hope that the revised manuscript is acceptable for publication in *Nature Communications*.

Reviewers' comments:

Reviewer #1 (Remarks to the Author):

The changes have improved the manuscript dramatically. The figures and movies are also much improved and in general very clear, especially considering the complexity of some of the items being illustrated.

I have remaining concerns with the descriptions and interpretations. Two fundamental issues are likely at the root of these problems.

First, the authors still try to describe their model using macroscopic analogies such as “squeezing” and “contraction” that are inappropriate at the molecular scale: matter is essentially incompressible at the molecular scale.

Second, the two new structures are both “apo” structures with respect to the transported substrate. Thus, it is likely that the differences between the two structures more accurately suggest the transition path between the outward-facing state after substrate release to the inward-facing state before substrate binding, rather than the transition that the authors seem most interested in – going from inward-facing to outward-facing in the presence of substrate followed by substrate release.

I believe Reviewer 1 had similar concerns when stating “In addition, I am not convinced by the model of extrusion presented. There may exist multiple OF intermediates and without a mechanistic context one can imagine a multitude of mechanisms.” In other words, the transition from inward- to outward-facing could well include outward-facing states with large substrate-containing cavities. These might be less stable in the absence of substrate (after the substrate unbinds and diffuses outward), leading to a high probability of transitioning to the observed crystalline state with a contracted chamber. Such an order of events makes more sense in microscopic terms, effectively generating a Brownian ratchet that insures substrate is released and the transport cycle continues.

To rephrase this last point: recurring in the manuscript is the idea of “squeezing/extruding/pumping” that seems to be used in the macroscopic sense. If this is

taken literally, it is then reasonable to ask whether a measurement of a physical force on the substrate can be made as it is extruded. In contrast, with microscopic reversibility and a Brownian ratchet there is no directed force on the molecule, but the equilibrium corresponding to the substrate binding and unbinding in some outward-facing state leads to a dissociation of the substrate and an outward-facing state that relaxes into a low-affinity apo state (which likely corresponds to the one observed in the crystal structure).

Reply: We thank the reviewer for the kind suggestion, which led to the further improvement of the paper. We generally agree with the recommendation that we moderate our claim in regard to two fundamental issues, macroscopic analogies vs. microscopic reversibility. Accordingly, we addressed all specific comments raised by the reviewer to meet this request to the greatest extent possible. Specifically, we removed the description about macroscopic analogies, and moderated the tone of our claim about the mechanism of transport, which was not supported by appropriate evidence.

Specific comments:

1. The revised title is still inappropriate, considering the comments above.

Reply: We agree with the reviewer. We previously considered the mechanism of substrate transport in greater detail, but in this manuscript we focused on the high-resolution structural information, clarified our explanations, and included the term ‘homodimeric P-glycoprotein’ in the new title:

“Inward- and outward-facing X-ray crystal structures of homodimeric P-glycoprotein CmABCB1”

2. Lines 10-14 of the abstract should be revised considering the comments above.

Reply: We agree with the reviewer, and rewrote the indicated sentences as follows:

“a newly identified chamber that significantly changes in volume with the aid of tight connections among transmembrane helices (TM) 1, 3, and 6; a glutamate–arginine interaction that stabilizes the outward-facing conformation; and strong interactions between TM1 and TM3, a property that distinguishes multidrug transporters from floppases. These structural elements are proposed to participate in the mechanism of the transporter.” (page 3, lines 11–15)

3. P. 4 line 16: “two types of inner gates” is rather confusing; “inner” is often used to refer to intracellular side of the membrane, for example, but here it seems to mean “internal”? Perhaps simplifying to “two gates” makes more sense?

Reply: We agree with the reviewer, and corrected “two types of inner gates” to “two gates.” (page 4, line 17)

4. P. 4 line 9: *ref(s) needed about side chains involved in substrate and inhibitor binding.*

Reply: We agree with the reviewer, and added the following reference:

Aller, S. G. *et al.* Structure of P-glycoprotein reveals a molecular basis for poly-specific drug binding. *Science* **323**, 1718-1722 (2009). (page 5, line 1)

5. p. 5 lines 8-9: *It's unclear to me what the authors mean by "which reaches beyond the intracellular membrane".*

Reply: We appreciate this comment. We rewrote the description to clarify our intended meaning as follows:

"which reaches the intracellular region beyond the cell membrane." (page 5, lines 9–10)

6. p. 8 lines 10-11: *"unwound and relaxed" and "regular and rigid" – the first qualifier is structure-based and makes sense. But the second one ("relaxed" and "rigid") requires additional evidence. An unwound region is not necessarily "relaxed", and "rigid" similarly is not warranted, unless there is evidence such as high or low B-factors in comparison to other elements in the structure, etc. I suggest simplifying the sentence by simply using "unwound" and "regular".*

Reply: We agree with the reviewer, and corrected the descriptions "unwound and relaxed" and "regular and rigid" to "unwound" and "regular", respectively. (page 8, lines 9–10)

7. p. 9 lines 12-14: *The description includes a time or order-of-events component that hasn't been demonstrated (twisting of TM6 "induced by" tilting of TM5; "Simultaneously"). The structures make it clear that all these structural changes do happen to reach the observed conformation, but the order in which they happen cannot be inferred simply from looking at the structure.*

Reply: We agree with the reviewer, and moderated the tone of our description of the temporal and order-of-events components as follows:

"which could be induced by tilting of TM5 or TM5*. Concomitantly, the inter-subunit TM6–TM1* or TM6*–TM1 interactions form via the same side chains (Fig. 2, upper panels)." (page 9, lines 12–13)

8. p. 12 lines 11-12: *"tension generated by the spring-like movement"? This is again a prediction that is disguised as an inference. It could be ok to edit to "and we propose that this deformation, a spring-like movement, generates tension to promote..."*

Reply: We agree with the reviewer, and corrected the description as proposed:
“We propose that this deformation, a spring-like movement, generates tension to promote...”.
(page 10, lines 12–14)

9. p. 13 line 8: this sentence seems supported only by Supp Fig. 4B, not 4A.

Reply: We thank the reviewer for raising this point. We moved the citation of Suppl. Fig. 4A from the previous sentence to the current sentence (p. 12 line 15).

10. p. 14 lines 7-9: I appreciate that the authors focus on analyzing sequences of transporters for which there are available structures to making inferences about specific structural features and the nearby sequence features, such as the contacts observed in TM1, TM3, TM6. However, to state that a given residue is conserved “across the B-subfamily ABC transporters” based on only this very limited and biased alignment is inappropriate (although it may be accurate, the evidence presented in Supp Fig. 3 is not enough to support it). The authors should either rephrase (these contacts are conserved in the transporters whose structures have been elucidated, which is the scope of the alignment) or use an alignment that more completely and deeply samples the B-subfamily transporters.

Reply: We agree with the reviewer, and corrected the description as follows and also added the description in the legend of Supp Fig.3:

"The alignment reveals that the residues participating in the Mg²⁺-nucleotide–NBD and NBD–TMD interactions are conserved in transporters whose structures have been elucidated (Supplementary Fig. 3)." (page 14, lines 11–13)

11. p. 14-15: the description of the proposed mechanism should be edited to address the above comments about macroscopic analogies vs. microscopic reversibility, and about which conformational transitions are most likely suggested by comparison of their two structures. Particularly, but not only, p. 14 lines 14-15, p. 15 lines 12-19. Also, the final sentence of the manuscript, p. 18 lines 3-4.

Reply: We agree with the reviewer. Accordingly, we removed the following description of the proposed mechanism in order to moderate the tone of our claim.

p.14 line 14 –p.15 line 1 (in the previous manuscript)

“, in which the drug is squeezed out by the inner chamber contraction. This contractile behavior of the chamber module is capable of squeezing out hydrophobic substrates with diverse chemical structures. The contraction would also promote uni-directional transport by preventing the substrate from returning from the outer space to the inner space of the transporter. Our high-resolution X-ray structures, especially the 1.9 Å outward-facing structure, reveal the roles of several conserved amino-acid residues in chamber contraction,

outer and inner gating, and communication between the NBDs and TMDs.”

p.15 line 5–8 (in the previous manuscript)

“In the inward-facing conformation, a hydrophobic substrate²³ enters via the gate opened toward the inner leaflet of the membrane bilayer between TM4 and TM6, and then moves to the top of the contractile chamber and interacts with the hydrophobic and aromatic residues on the “ceiling”⁹.”

p.15 line 18–p.16 line 1 (in the previous manuscript)

“We speculate that the squeezing motion of the TMs constituting the chamber lifts the substrate, facilitating opening of the extracellular gate by a wedge-like movement of the substrate.”

p. 18 lines 3-4. (in the previous manuscript)

The strain/release motion of transmembrane helices progressively contracts the inner chamber to squeeze a large variety of chemicals out of the cell.

12. p. 16 lines 6-8: the conformation of the transporter in an outward-facing state with bound substrate (although it may be challenging to observe depending on how stable it is) is needed to make a statement about this. For all we know, there is a transient state with openings to the lipid bilayer.

Reply: To moderate the tone of our claim, we rewrote the sentence as follows:

“The narrowness of the opening of the extracellular gate in our structure of the outward-facing state suggests that substrates are transported to the extracellular space rather than into the lipid bilayer (Supplementary Fig. 2A).” (page 15, lines 17–19 to page16, line 1)

13. Table 1: The overall CC1/2 value is missing for the Hg derivative.

Reply: Because the Hg derivative dataset was processed by using an older version of HKL2000 (ver. 0.98), the overall CC1/2 value has not been calculated. We mention this in the footnote of Table 1.

14. In Supp. Fig. 1 and Supp. Fig. 5, ATPase activity assays are done at (n = 2) and SD means very little when using only two measurements. In such a situation, it is generally more prudent to report the two measurements as replicates (plotted as two points, instead of bars/points with error bars) without using mean +/- SD.

Reply: Thank you for the suggestion. Accordingly, we redrew the Figures (Supp. Fig. 1 and Supp. Fig. 5).

Reviewer #2 (Remarks to the Author):

The manuscript by Kodan et al. reports two crystal structures of a eukaryotic homolog of Pgp, CmABCB1. One of these structures is referred to as outward-facing which represents the first such structure of a Pgp homolog at high resolution. Starting from their previous inward-facing structure of this protein, the authors designed a mutant (QTA) which stabilize the OF. In addition to describing the structure, the manuscript contains instructive comparisons with bacterial ABC exporters as well as recently determined cryoEM structure of human Pgp. While the latter structure somewhat diminishes the impact of this work, there are atomic details that provide insights into the stability of this state as well as the mechanism of alternating access.

Reply: We very much appreciate the reviewer's positive evaluation of our work and its significance.

The current version of the manuscript is much improved with the speculative aspects toned down. But there are still concerns that the authors need to address:

Reply: Below, we provide our point-by-point responses to the concerns raised by the reviewer.

1- In both title and abstract, the authors must note that this is a eukaryotic homolog of Pgp. This is important in light that this is a homodimeric ABC transporter.

Reply: We agree with the reviewer that CmABCB1 is homodimeric, and that this feature is distinct from mammalian Pgps. Therefore, we changed the title and revised the related sentence in the manuscript as follows:

Title: "Inward- and outward-facing X-ray crystal structures of homodimeric P-glycoprotein CmABCB1"

In abstract: "Here we report a pair of structures of homodimeric P-glycoprotein: ..." (page 3, lines 7–8)

"In general, the outward-facing high-resolution crystal structure of Mg²⁺-nucleotide-bound homodimeric CmABCB1 is similar to the cryo-EM structure of monomeric human P-gp¹⁵, ..." (page 16, lines 2–5)

2- The authors suggest that the double mutation of Q147ATM1/T381ATM6 "had minimal effect on protein function of drug resistance and ATPase hydrolysis activity." In another sentence they claim "...the QTA mutation only affected the conformational equilibrium between inward-facing and outward-facing states." This is mutually exclusive. If the energy

landscape is modified then there has to be a functional consequence. Indeed in supplementary Figure 1, the results show the QTA mutant has a higher stimulated ATPase activity (more than 30%) in the presence of Rhodamine and a lower maximal ATPase hydrolysis (30% less) compared with the wild-type CmABAB1. This is inconsistent with the authors' statement. The fact that the mutant may have minimal impact on drug transport in vivo does not suggest unaffected mechanism of function in vitro, as clearly demonstrated here. Rather than shying away from this result, the authors should put it in mechanistic context.

Reply: We agree with the reviewer that the two sentences are mutually exclusive.

Accordingly, we removed the latter sentence "...the QTA mutation only affected the conformational equilibrium between inward-facing and outward-facing states."

3- A cluster of residue referred to as the "chamber contraction regulator" are proposed to stabilize the OF. First, I do not see a reason for the term particularly that it is speculative. The experiments to test the role of these residues involve mutations to alanine. While the substitutions do affect transport, they are not specific to the mechanism proposed. In fact the drastic changes render these experiments of limited value. The interpretation should be toned down.

Reply: Thank you for this comment. We want to emphasize that there is experimentally determined high-resolution structural evidence. The structural results clearly showed that the atomic distances and stereochemical coordinates are suitable for close inter-subunit contacts, especially via the van der Waals contacts of Gln398^{TM6}-Ala240*^{TM3}, and that they stabilize the contracted chamber in the outward-facing structure (Fig. 3D), whereas no such contacts are observed in the inward-facing structure with a large internal cavity (Fig. 3C). Indeed, the biochemical data for the mutants are not specific to the mechanism, but do not conflict with the structural interpretation. Therefore, we wish to propose a possible function based on the structural evidence, and preserve the term "chamber contraction regulator." According to the reviewer's suggestion, we moderated our claim as follows:

"Because these residues seem to regulate the chamber space, we refer to them collectively as the 'chamber contraction regulator'." (page 10, lines 2-3)

4- On a more fundamental level, what is the basis of referring to the structure as OF? In fact the opening to the outside is limited as noted by the authors. This is strikingly illustrated with the comparison to the two bacterial homodimers. Both this structure and human Pgp are less open and have diffuse density in this area. The manuscript suggests that this may be an intrinsic mechanistic difference between multidrug transporters and floppases. An alternative interpretation is that these structures may reflect a mostly occluded conformations prior to the OF state. In fact the DEER study cited by the authors propose the presence of this intermediate. The authors should discuss this possibility.

Reply: Thank you for giving us the opportunity to re-evaluate this point. We do consider, however, that the current Mg²⁺-AMP-PNP-bound QTA structure is a kind of OF state, and is not likely to be an occluded conformation because the contracted chamber within the TMD region has no space for the bound substrate, as shown in Fig 1D & H. The DEER study did not obtain experimental evidence for the chamber region, and their proposed mechanism including the occluded state is not supported by experimental evidence. Hence, it is difficult to discuss the occluded state based on our structures without indulging in speculation. Instead, we emphasized that this structure suggests an intrinsic mechanistic difference between multidrug transporters and floppases, as proposed by the reviewer. The phrase “a property that distinguishes multidrug transporters from floppases.” was added to the abstract (line 14), and the sentence “, suggesting an intrinsic mechanistic difference between multidrug transporters and floppases” was added on page 12, line 9.

5- There are a number of loosely phrased, partially inaccurate statements that require attention

a- In the abstract, the statement that "lack of structural information regarding other conformational states..." is not accurate. There is now a structure of Pgp bound to an inhibitor and one in a putatively OF conformation.

Reply: We agree that there are several types of inward- and outward-facing conformations. We revised the sentence as follows:

“the lack of high-resolution structural information regarding the alternate conformational states of the molecule.” (page 3, lines 4–7)

b- abstract needs to be rewritten to make a crisp distinction between mammalian Pgp and the homodimer under study in this manuscript

Reply: According to the reviewer’s suggestion, to distinguish between mammalian Pgp and homodimer CmABC1, we now use the term “homodimeric P-glycoprotein” in the abstract:

“Here we report a pair of structures of homodimeric P-glycoprotein: ...” (page 3, lines 7–8)

c- P4 line 12 “ no outward facing crystal structure” but there is a cryoEM structure

Reply: Although they are both structures, we make a distinction between crystal structures and cryoEM structures due to differences in precision and reliability. The cryoEM human P-gp structure is low-resolution, and it is difficult to define the side chain positions and bound ligands. Therefore, to clarify our intention, we simply changed the description as follows: “no outward facing X-ray crystal structure.” (page 4, line 12)

d- page 4 line 17: the use of the word “gates” imply a specific mechanism, please change.

Reply: We previously described the “gates” as follows: “These structures, along with site-directed mutagenesis and transporter activity measurements, reveal the detailed architecture of the transporter, including a gate that opens to extracellular side and two gates that open to intramembranous region and the cytosolic side.” (Kodan, A. *et al. Proc Natl Acad Sci U S A* **111**, 4049–4054 (2014)). We referred to this paper in this sentence. (page 4, lines 18)

e- page 6, line 12 “this ensureshydrolysis”. This statement is ambiguous. How does a free energy process ensure a specific rate? How is the return to inward facing conformation following ATP hydrolysis irreversible?

Reply: We agree that this sentence was ambiguous, so we removed it.

f- Page 13 line19 : the role of the RE latch is speculative. The sentence starting from RE-latch and ending online 1 page 14 should be deleted. No information on kinetics is available in this manuscript.

Reply: We agree with the reviewer. Accordingly, we removed the sentence “RE-latch seems to transiently stabilize the NBD dimer to allow time for the export of hydrophobic substrates before ATP hydrolysis”. In addition, we suggest a role for the RE-latch in the section discussing the proposed mechanism, as follows:

“It is possible that the opening of the extracellular gate liberates the substrate to the extracellular space, and that the RE-latch (Glu620 and Arg644) stabilizes the outward-facing conformation until the substrate is expelled.” (page 15, lines 12–14).

g- I find the section on mechanism of transport too long. The author should summarize their findings crisply stating clearly that what they have is a structural mechanism of alternating access and not transport. I find the discussion about how the drug is exported speculative because of the lack of an intermediate state structure that informs on how the substrate threads its way to the outward-facing cavity. MsbA and Pgp have IF and OF structures. The only added insight in this manuscript comes from the high resolution and the ability to discern subtle structural aspect. The authors are encourage to avoid excessive speculations on how substrates are pushed through.

Reply: We agree that the section on the mechanism of transport was too long. However, we believe that the discussion of this topic is important and significantly advances the reader’s understanding of the mechanistic rationale for the transport of multiple drugs by the dimeric P-gp CmABCB1. Hence, we removed the following sentences to moderate the tone of our claim regarding how the drug is exported, and to shorten the section.

p.14 line 14 –p.15 line 1 (in the previous manuscript)

“, in which the drug is squeezed out by the inner chamber contraction. This contractile behavior of the chamber module is capable of squeezing out hydrophobic substrates with diverse chemical structures. The contraction would also promote uni-directional transport by preventing the substrate from returning from the outer space to the inner space of the transporter. Our high-resolution X-ray structures, especially the 1.9 Å outward-facing structure, reveal the roles of several conserved amino-acid residues in chamber contraction, outer and inner gating, and communication between the NBDs and TMDs.”

p.15 line 5–8 (in the previous manuscript)

“In the inward-facing conformation, a hydrophobic substrate²³ enters via the gate opened toward the inner leaflet of the membrane bilayer between TM4 and TM6, and then moves to the top of the contractile chamber and interacts with the hydrophobic and aromatic residues on the “ceiling”⁹.”

p.15 line 18–p.16 line 1 (in the previous manuscript)

“We speculate that the squeezing motion of the TMs constituting the chamber lifts the substrate, facilitating opening of the extracellular gate by a wedge-like movement of the substrate.”

p. 18 lines 3-4. (in the previous manuscript)

The strain/release motion of transmembrane helices progressively contracts the inner chamber to squeeze a large variety of chemicals out of the cell.

Reviewers' Comments:

Reviewer #1:

Remarks to the Author:

The authors have satisfied my previous requests and in the process, improved the clarity and scientific accuracy of the manuscript.

One minor remaining suggestion. In their revised abstract on lines 13-14, they state "and strong interactions between TM1 and TM3". The word choice is confusing because they do not measure forces or energies. I believe they mean "extensive interactions" instead.

Response to Reviews

We are pleased that the reviewers approved of the improvements in the revised version of the manuscript. We have addressed a concern raised by reviewer #1 in the revised manuscript.

Reviewers' comments:

Reviewer #1 (Remarks to the Author):

The authors have satisfied my previous requests and in the process, improved the clarity and scientific accuracy of the manuscript.

One minor remaining suggestion. In their revised abstract on lines 13-14, they state "and strong interactions between TM1 and TM3." The word choice is confusing because they do not measure forces or energies. I believe they mean "extensive interactions" instead.

Reply: We agree with the reviewer, and corrected the descriptions “strong interactions” to “extensive interactions” in abstract on line 11.